# Molecular Insights into the Role of Pathogenic nsSNPs in *GRIN2B* Gene Provoking Neurodevelopmental Disorders

**DOI:** 10.3390/genes13081332

**Published:** 2022-07-26

**Authors:** Abid Ali Shah, Marryam Amjad, Jawad-Ul Hassan, Asmat Ullah, Arif Mahmood, Huiyin Deng, Yasir Ali, Fouzia Gul, Kun Xia

**Affiliations:** 1Center for Medical Genetics and Hunan Key Laboratory of Medical Genetics, School of Life Sciences, Central South University, Changsha 410013, China; alishah.abid666@gmail.com (A.A.S.); khanbiochemist007@gmail.com (A.M.); 2District Headquarter (DHQ) Hospital, Faisalabad 38000, Punjab, Pakistan; marryamamjad193@yahoo.com; 3Allied Hospital, Faisalabad 38000, Punjab, Pakistan; dr.jawad386@gmail.com; 4Novo Nordisk Foundation Center for Basic Metabolic Research, Faculty of Health and Medical Sciences, University of Copenhagen, 2200 Copenhagen, Denmark; asmatullah@bs.qau.edu.pk; 5Department of Anesthesiology, The Third Xiangya Hospital of Central South University, Changsha 410013, China; huiyind@umich.edu; 6National Center for Bioinformatics, Quaid-i-Azam University, Islamabad 45320, Pakistan; yasirkhanqu@gmail.com (Y.A.); fouziagul97@gmail.com (F.G.); 7Hengyang Medical School, University of South China, Hengyang 421000, China; 8CAS Center for Excellence in Brain Science and Intelligences Technology (CEBSIT), Chinese Academy of Sciences, Shanghai 200030, China

**Keywords:** *GRIN2B*, neurodevelopmental disorders, molecular dynamics simulation, developmental delay, Lennox–Gastaut syndrome, SNPs

## Abstract

The GluN2B subunit of N-methyl-D-aspartate receptors plays an important role in the physiology of different neurodevelopmental diseases. Genetic variations in the GluN2B coding gene (*GRIN2B*) have consistently been linked to West syndrome, intellectual impairment with focal epilepsy, developmental delay, macrocephaly, corticogenesis, brain plasticity, as well as infantile spasms and Lennox–Gastaut syndrome. It is unknown, however, how *GRIN2B* genetic variation impacts protein function. We determined the cumulative pathogenic impact of *GRIN2B* variations on healthy participants using a computational approach. We looked at all of the known mutations and calculated the impact of single nucleotide polymorphisms on *GRIN2B*, which encodes the GluN2B protein. The pathogenic effect, functional impact, conservation analysis, post-translation alterations, their driving residues, and dynamic behaviors of deleterious nsSNPs on protein models were then examined. Four polymorphisms were identified as phylogenetically conserved PTM drivers and were related to structural and functional impact: rs869312669 (p.Thr685Pro), rs387906636 (p.Arg682Cys), rs672601377 (p.Asn615Ile), and rs1131691702 (p.Ser526Pro). The combined impact of protein function is accounted for by the calculated stability, compactness, and total globularity score. GluN2B hydrogen occupancy was positively associated with protein stability, and solvent-accessible surface area was positively related to globularity. Furthermore, there was a link between GluN2B protein folding, movement, and function, indicating that both putative high and low local movements were linked to protein function. Multiple *GRIN2B* genetic variations are linked to gene expression, phylogenetic conservation, PTMs, and protein instability behavior in neurodevelopmental diseases. These findings suggest the relevance of *GRIN2B* genetic variations in neurodevelopmental problems.

## 1. Introduction

N-methyl-D-aspartate receptors (NMDARs or GluNRs), which are excitatory glutamate receptors expressed widely in the brain, perform crucial roles in neuronal processes such as neurodevelopment, synaptogenesis, plasticity, learning, and memory [1]. This larger heteromeric complex is made up of seven *GRIN* genes (*GRIN1*, *GRIN2A-D*, and *GRIN3A-B*) encoding subunits for N-methyl-D-aspartate-type glutamate receptors, respectively. One among these is *GRIN2B*, which provides instructions for making a protein called GluN2B [2]. This gene is involved in a variety of neurodevelopmental diseases such as West syndrome and intellectual impairment with focal epilepsy [3] and may be linked to infantile spasms and Lennox–Gastaut syndrome [4]. *GRIN2B* variations have been identified in *GRIN2B*-related neurodevelopmental disorders, such as ID [5], ASD [6,7], Atypical Rett Syndrome & intractable epilepsy [8], encephalopathy [9], developmental delay or macrocephaly [10], schizophrenia [11], obsessive compulsive disorder [12], and cerebral visual impairment [13], causing cells to create an inactive GluN2B protein or prevent them from making any GluN2B protein. Researchers are uncertain how improper NMDA receptor activation compromises typical brain growth and development or why excessive or inadequate activity manifests in neurological complications in people with *GRIN2B*-related neurodevelopmental disorders.

The three-dimensional structure of GluN2B includes conserved domains of the NR2B subunit, i.e., signal peptide (SP) from the residue (1–28), amino-terminal domain (ATD) (28–393) that is involved in receptor assembly, S1 (458–547), S2 (681–798) that forms the ligand-binding domain, re-entrant pore-forming and transmembrane spanning domains (Pore) (556–681) and the C-terminus PDZ domain-binding motif (PDZ) (840–1484), respectively. Disease associated *GRIN2B* variants have been reported in N-terminal domain (NTD), membrane-proximal ligand binding domain (LBD), transmembrane domain that forms the ion pore containing three transmembrane (TM) spanning domains followed by an intracellular C-terminal PDZ-binding domain (CTD) of variable length [14]. Mutations arising due to West syndrome were found in the re-entrant pore-forming domain, while ID and focal epilepsy mutations were found in the glutamate-binding domain S1 [3]. In recent times, SNPs in coding and regulatory areas were employed as markers in association and linkage studies to determine the region of the genome associated with a certain disease [15,16,17,18,19]. Because each variation has the potential to modify a protein’s function or structure, SNP detection studies and mutagenesis analysis are carried out together to uncover amino acid variations in protein-coding domains [20]. Understanding a major proportion of disease-causing genetic mutations is a key aim in today’s genetics [21]. A characterization of variations based on their nature, large-scale investigations of SNPs, and in-depth association studies are all necessary for linking a gene to a certain disease [22]. More sophisticated computational approaches are being developed these days to aid in the development of a high-throughput, practical method for detecting changes in protein structure, capacity, and strength as a consequence of a change [23,24,25].

The current study sought to identify and forecast pathogenic nsSNPs in the *GRIN2B* gene, as well as their associations with disease and the influence of deleterious nsSNPs on protein structural behavior. Different bioinformatics tools such as SIFT [23], Polyphen2 [26], SNAP2, SNPs&GO [27], PhD-SNP [28], I-Mutant [29], and Mu-Pro [30] were used to predict pathogenic nsSNPs in the *GRIN2B* gene. In addition, ConSurf [31] was used to detect amino acid residue conservation. Along with these, MusiteDeep and Findmod were used to identify the protein’s post-translational modification sites [32]. AlphaFold2 [33] was used to generate three-dimensional structures of mutant and natural PDS proteins, which were further refined using ModRefiner [34]. To explore the structural variety and changes in interaction behaviors of natural and mutant proteins, the COACH server was used to predict ligand-binding sites [35]. Through protein-ligand docking, PyMOL [36] was used to examine how natural and mutant proteins interact with their environment. Schrodinger was used to execute molecular dynamics simulations and visualize structural variations over time. The detailed visualization analysis was done using PyMol [36].

## 2. Materials and Methods

### 2.1. Data Collection

NCBI (http://www.ncbi.nlm.nih.gov, accessed on 1 April 2022) was used to get the nucleotide sequence in FASTA format [37], as well as the Glutamate receptor ionotropic protein amino acid sequence (Q13224) from the UniprotKB database (https://www.uniprot.org/, accessed on 01 April 2022) [38]. SNPs of the *GRIN2B* gene were found in the NCBI database of SNPs (dbSNP (http://www.ncbi.nlm.nih.gov/snp, accessed on 1 April 2022) [39], and Ensemble genome browser (https://www.ensembl.org, accessed on 1 April 2022) [40]. OMIM database (http://www.omim.org, accessed on 1 April 2022) was used to find information on the Glutamate receptor ionotropic gene and protein [41].

### 2.2. Prediction of SNP Functional Effect

SIFT [23], Polyphen2 [42], and SNAP2 (https://www.rostlab.org/servces/snap, accessed on 5 April 2022) [43] online tools and servers were used to anticipate the effects of SNPs. Both servers consider a variety of inputs and categorize potential SNPs based on their respective confidence ratings. SIFT assigns a tolerance index score to each mutation, with a value of 0.05 deemed detrimental. The PolyPhen2 score, on the other hand, ranges from 0 to 1, with zero suggesting that amino acid alterations do not affect protein function and 1 indicating the most harmful behavior. SNAP2 allows for genome-wide comparisons and predicts the functional impact of amino acid alterations. The common nsSNPs from both servers were chosen for further study.

### 2.3. Prediction of Disease Association of SNPs

PhD-SNP (http://snps.biofold.org/phd-snp/phd-snp, accessed on 6 April 2022) [28] and SNPS&GO (http://snps-andgo.biocomp.unibo.it/snps-andgo/snps-andgo/snps-andgo, accessed on 6 April 2022) [27] were used to assess the association of filtered SNPs with disease pathogenesis. PhD-SNP is an online program that predicts the association between illnesses and SNPs with a 78 percent accuracy rate. SNPs are categorized as illness-related or neutral and graded from 0 to 9. SNPs & GO is a dependable approach that predicts disease-related amino acid changes in protein and functional categories at a single site, with an overall prediction accuracy of 82%. SNPs & GO was given the UniProt accession number (Q13224) of Glutamate receptor ionotropic protein as well as the mutation site of both native and changed amino acids as inputs. CADD [44], REVEL [45], MetaLR, and Mutation Assessor [26] were employed concurrently to validate the pathogenic effect of these changes to prevent false positives in the data.

### 2.4. Impact on Protein Stability

SNPs affect protein strength, which may either reduce or improve protein stability. Several strategies were used to boost the confidence in the changes produced to foresee these impacts. I-Mutant3 (http://folding.biofold.org/i-mutant/i-mutant3.0, accessed on 10 April 2022) [29] is software that predicts how SNPs will affect the stability of a protein. The tool’s accuracy might reach 77%. I-Mutant was fed the GluN2B protein amino acid sequence as well as mutations of residues and their positions. MUpro, a machine learning system collection, identifies changes in protein states and strength caused by amino acid mutations [30]. The inputs for MUpro were identical to those for I-Mutant, with the exception that MUpro supports substitution positions in addition to original and changed residues.

### 2.5. Sequence Conservation Analysis

ConSurf, an online software (http://ConSurf.tau.ac.il/, accessed on 10 April 2022), was used to analyze GRIN2B protein conservation. ConSurf is a useful software for predicting protein target area high-throughput functions [31]. For each protein residue of relevance, the conservation analysis is shown on a scale of 1 to 9. On the scale, 1–3 represent variable, 4–6 represent the average, and 7–9 represent highly preserved regions [31]. The tool takes FASTA protein sequences as input.

### 2.6. Post-Translational Modifications Sites (PTMs) Prediction

A variety of amino acid alterations occur at post-translational modification sites, resulting in the creation of a varied spectrum of proteins. PTM sites have been found, including methylation, phosphorylation, acetylation, and ubiquitination. These sites play important roles in cellular architecture, such as protein-protein interactions and disease-related signaling cascades. As a consequence, projecting PTM data assists in assessing the influence of polymorphisms on disease association or pathogenicity. The total PTMs were predicted using MuSiteDeep (https://www.musite.net/, accessed on 16 April 2022) [32]. MuSiteDeep uses protein sequences to predict PTMs. It is made up of 34 strategic relapse abstractions that were constructed individually from a set of 126,036 non-excess realistically verified sites for 23 distinct varieties using accessible datasets. Later, Findmod was also used to identify PTM-contributing peptides (http://www.expasy.ch/sprot/findmod.html, accessed on 20 April 2022).

### 2.7. Homology Modelling

The structural stability of the original and mutant proteins was assessed using structural analysis. Because the full-length GluN2B protein structure has yet to be determined, we predicted the native protein structure using the AlphaFold software [33], which is a unique machine learning strategy that combines physical and biological knowledge about protein structure into the construction of the deep learning algorithm by exploiting multi-sequence alignments. SPDViewer was used for mending atoms nomenclature and modeling sidechains [46], YASARA was used to obtain the lower energy minima of the modeled structure [47], and the “swappaa” feature of UCSF Chimera was used for mutant models production, steric clashes were removed and the mutant models were minimized for 1500 steps (750 steepest descent and conjugate gradient) [48]. The SAVES server (http://servicesn.mbi.ucla.edu/SAVES, accessed on 23 April 2022) was used to validate protein models. It contains six integrated modules, one of which is the RAMACHANDRAN plot, which is used for protein structural assessment and offers information on the conformation of residues in authorized and banned positions.

### 2.8. Molecular Dynamics Simulations

The Desmond program of Schrödinger software 2021-2 (Schrödinger, LLC, New York, NY, USA) [49] with the OPLS4 (Optimized Potentials for Liquid Simulation) [50] force field was used for MD simulation to examine the conformational changes in proteins’ dynamic movements. A simulated triclinic periodic boundary box with a 10 extension from each direction was developed to solve the GluN2B protein structures, and an explicit solvation model (Monte-Carlo equilibrated SPC, the transferable intermolecular potential of 3 points) was utilized for each system. Lennard-Jones interactions (cut-off = 10) and the SHAKE algorithm were employed to govern the mobility of all covalent bonds, including hydrogen bonds [51]. During the solvation, additional counter ions (0.15 M Na + Cl) were used to neutralize the whole system. At 300 K and 1 bar pressure, the protein models were subjected to energy reduction until a gradient threshold of 25 kcal/mol/ was achieved using the NPT ensemble class. Each system had a single MD run, and the trajectory was recovered by setting the time period for all simulated trajectories to 20 ps. The Particle Mesh Ewald (PME) algorithm was used to calculate long-range coulombic interactions, and the RESPA integrator (a motion integration software) [52] was used to control all covalent bonds coupled with hydrogen atoms, with an inner time step of 2 fs throughout the simulation. For short-range electrostatic interactions, a cut-off of 9.0 was selected, whereas, for long-range van der Waals (VDW) interactions, a uniform density approximation was used. A Nosé–Hoover thermostat [53] with a relaxation duration of 12 ps was utilized at 300 K and 1 atmospheric pressure. The Martyna–Tobias–Klein barostat approach [54] with a relaxation length of 12 ps was used to maintain the condition throughout the simulation. The stability of each system was then assessed using RMSD (Root Mean Square Deviation), RMSF (Root Mean Square Fluctuation), Rg (Radius of gyration), H-bond occupancies, and SSE (Secondary Structure Elements) using the MD simulation trajectories via Schrödinger 2021-2.

### 2.9. Dynamic Cross-Correlation Map (DCCM)

A 3D matrix representation of the time-dependent migration of amino acid residues. This approach compares C atoms throughout the correlation matrix for all complexes and systems to discover the continuous correlations of domains. The DCCM investigation used 9800 images based on C carbon atoms. The displacements of the backbone C atoms in the trajectories were found to be cross-correlated with the displacements of the C carbon atoms. DCCM has a value between 1 and +1, where a value larger than zero denotes movement with positive correlation (same direction) between two atoms, and a value less than zero represents movement with negative correlation (opposite direction). ProDy was used to evaluate the DCCM [55], and Matplotlib was used to plot the results [56]. The DCCM graph illustrates two types of correlation: positive and negative correlation. A positive correlation indicates that the ligand and protein movements are parallel, and that the system is stable when they interact. A negative correlation, on the other hand, indicates that the ligand is migrating out of the binding pocket, resulting in an anti-parallel correlation or that the complex is unstable. Furthermore, the intensity of the colors in the DCCM map is related to the strength of the positive and negative correlations. These connections are represented by the colors red to light red and blue to light blue. The red color represents positive correlation, while the blue color represents negative correlation; a deeper color suggests a more meaningful link and vice versa.

### 2.10. PCA Analysis

PCA analysis using ProDy was used to analyze the major movements of high amplitude [55]. The diagonalized covariance matrix was used to create the covariance matrix, which was then used to analyze the eigenvalues and eigenvectors. The PCs, or eigenvectors, reflect the direction of movement of the ligand and receptor atoms, while the associated eigenvalues describe the complex’s mean square fluctuations. PC1 and PC2 were utilized for calculating and graphing to verify their movements.

## 3. Results

### 3.1. Data Acquisition and SNP’s Annotation

The human *GRIN2B* gene has a total of 1963 single nucleotide polymorphisms that were annotated based on mutation type and loci. There were 838 missense mutations, 623 synonymous mutations, 312 stop gain mutations, and 96 in the protein’s splice region. There were 39 coding sequence SNPs identified, with 28 known to be frameshifts and 12 to be in-frame deletions, 7 splice donors, and 6 splice acceptor variations. The stop codon was maintained in just two variants. Figure 1 displays the distribution of SNPs by category. 25 of the 39 nonsynonymous SNPs in the coding sequence were clinically significant.

### 3.2. Prediction of Nature of SNPs

To get high confidence findings, the SIFT and Polyphen2 servers were utilized concurrently to estimate the impact of SNPs. SIFT and Polyphen2 were fed a total of 25 clinically important nsSNPs. According to SIFT indexing, 24 (96 percent) of the clinically significant nsSNPs were projected to be deleterious with a confidence level of 0.00, while just one SNP was indexed as tolerated. PolyPhen2 which is indexed based on Multiple Sequence Alignment and structural information, classified 20 of the nsSNPs as “deleterious”, 3 as “potentially deleterious”, and just two as “benign”, summing up 25 nsSNPs to be harmful.

To get the most confident findings, the servers SNAP2, CADD, REVEL, MetaLR, and Mutation Assessor were utilized to filter out the most deleterious nsSNPs. These 25 nsSNPs were then sent to the SNAP2 server for validation. This study found that 24 of the nsSNPs had an “impact” on protein function, whereas just one was anticipated to be “neutral”. Table 1 is a depiction of the functional effects of nsSNPs. Six nsSNPs (p.Arg1111His, p.Gly826Glu, p.Ser810Arg, p.Gly689Ser, p.Val558Ile, and p.Asn516Ser) were evaluated as having no or little influence by two or more tools and were therefore removed from further investigation.

### 3.3. Prediction of SNPs Associated with Disease

The filtered and decreased numbers of polymorphisms were then tested to see whether they were disease-linked or not. The SNP & GO and PhD-SNP servers were then given 19 high-confidence nsSNPs. Each nsSNP was assigned a score to determine if it was neutral or disease-related. To generate high-confidence findings, a cut-off value (*p* ≥ 0.80) for PhD-SNP scores was used. Only four nsSNPs (p.Ile751Thr, p.Ilr695Thr, p.Asp668Asn, and p.Glu413Gly) were determined to be neutral by PHD-SNP, whereas the remaining 15 nsSNPs were disease-causing, as predicted by SNPs and GO. The filtered results are shown in Table 2.

### 3.4. Prediction of Effect of Stability of Protein

The I-Mutant and MuPro servers were used to estimate the structural impact of 19 potential nsSNPs, with I-Mutant predicting just three nsSNPs (p.Gly820Ala, p.Asn615Ile, and p.Ser526Pro) enhancing protein stability and the other 16 nsSNPs reducing protein activity by decreasing its stability. MuPro anticipated that none of the nsSNPs would increase protein stability. Table 3 gives a depiction of the findings. Because none of the nsSNPs were shown to be neutral by more than one technique, none of the SNPs were removed from further study.

### 3.5. Sequence Conservation Analysis

Any disease-causing mutation is usually discovered in a gene’s highly conserved region. The ConSurf server was used to investigate the conserved behavior of the 25 nsSNPs. On the scale, 9 nsSNPs were highly conserved in addition to being structural and buried, two were conserved and only buried, and five were predicted to be highly conserved in addition to being functional and exposed, while two were just exposed. The one that survived (p.Arg682Cys) was well preserved and exposed. The ConSurf results were represented as color codes ranging from blue to purple, with blue showing variability and purple indicating a highly conserved location. As shown in Figure 2, ConSurf projected that the variations would be buried (b) or revealed (e), as well as functional (f) or structural (s). The functional and structural implications of the 19 high confidence nsSNPs, as well as the PTM positions and phylogenetic conservation scores, are summarized in Table 3.

### 3.6. Prediction of PTMs (Post Translational Modifications) Sites

The MusiteDeep and FindMod servers predicted post-translational modifications linked with our candidate nsSNPs by using the protein sequence as input and an extra peptide mass as input for FindMod. The findings were obtained in .csv format, and all 19 nsSNPs were evaluated. Only three nsSNPs (p.Thr685Pro, p.Arg682Cys, and p.Asn615Ile) were shown to contribute to methylation, phosphorylation, and glycosylation, respectively. FindMod also discovered two nsSNPs (p.Arg682Cys and p.Asn526Pro) in peptides that contribute to 2,3-didehydroalanine (Ser) and bromination. Table 3 has the descriptions of the 19 SNPs that were either directly involved or contributed to the peptide that was implicated in PTMs. Table 4 summarizes the specifics of the mutations that were exposed to MD modeling.

### 3.7. Structure Prediction

Our findings show that mutations in the GluN2B protein at p.Thr685Pro, p.Arg682Cys, p.Asn615Ile, and p.Ser526Pro induce pathogenicity. We picked these four highly conserved mutations based on their prediction scores to investigate the protein structural changes they produce. Using AlphaFold, we obtained 10 templates, of which 7EU8.B exhibited 100 percent identity despite only covering 56 percent of the query sequence. This template served as the foundation for our model’s backbone and structure prediction. The comparison models were created using the Discrete Optimized Protein Energy (DOPE) evaluation score to differentiate between “good” and “poor” models. Figure 3 depicts the natural protein structure of GluN2B. After predicting the original protein structure using AlphaFold, mutant structures (GluN2B^p.Asn615Ile^, GluN2B^p.Thr685Pro^, GluN2B^p.Arg682Cys^, and GluN2B^p.Ser526Pro^) were constructed with UCSF Chimera using the “aaswap” command-line method. RAMACHANDRAN plot structure validation revealed that 93.32 percent of the anticipated native structure’s residues occupy space in the most preferred area, while 4.77 percent occupy space in the additionally favored region. The verified structures were subjected to molecular dynamics modeling.

### 3.8. Explicit Solvent Molecular Dynamics of Pathogenic Mutations

#### 3.8.1. Stability Analysis

The disruptive effect of GluN2B upon undergoing mutations was revealed by molecular dynamics simulations of 100 ns. Major structural changes were embraced by GluN2B^WT^ for the first third of the simulation, however, the RMSD tends to converge at 12 Å for the latter part. Detailed analysis of the three-dimensional structures revealed that during the first 20 ns, the major contributor to the conformational shift was loops which changed drastically during that period, while α-helices also contributed to the change in conformation. On the other hand, GluN2B^p.Asn615Ile^ was more stable at 9 Å for the first half of the simulation, undergoing some conformational changes before converging on the same. GluN2B^p.Thr685Pro^ got stable after 50 ns and remained stable till the end of the simulation. GluN2B^p.Arg682Cys^ embraced seasonal stability while exhibiting more fluctuation occasionally. GluN2B^p.Ser526Pro^ remained unstable till the end of the simulation time. The overall trend in the RMSD values was relatively more distant from the GluN2B^WT^, which shows that upon point mutation, the GluN2B experiences a change in stability (Figure 4), however, the changes were more prominent in all mutants except GluN2B^p.Asn615Ile^, which showed relatively comparable RMSD against GluN2B^WT^.

#### 3.8.2. Flexibility Comparison of Native and Mutant GluN2B Structures

The collective flexibility of all the systems was inferred from residual fluctuations of the GluN2B^WT^ and GluN2B^MT^ models. The RMSF scores of GluN2B^WT^ protein’s residues; Phe580 to Pro598 were noted above 10 Å with Asp591 exhibiting the highest of 13.41 Å, while other most fluctuated peaks were Arg847 to Pro858 and Pro870 to Ile872 also exhibited RMSF of more than 10 Å. Other small RMSF peaks were Asn444 (8.03 Å) and Thr626 (9.99 Å).

On the other hand, the RMSF of GluN2B^p.Asn615Ile^ suggested that Cys588-Pro595 and His848- Ser865 exhibited more than 10 Å with Gly592 posing 11.92 Å and Val853 posing 13.94 Å. The other fluctuating peaks were Asp447 (6.69 Å), Thr626 (8.74 Å), and Asn806 (6.33 Å). The scores of GluN2B^p.Thr685Pro^ suggested that Phe577-Gly597 and His840-Ile872 exhibited more than 10 Å with Arg593 posing 14.68 Å and Cys854 posing 19.06 Å. The other fluctuating peaks were Leu209 (7.96 Å), Asp447 (7.73 Å), and Gly626 (11.10 Å). The RMSF of GluN2B^p.Arg682Cys^ suggested that Ser810 and Arg593 exhibited the highest RMSF of 9.69 Å and 9.01 Å. The flexibility of GluN2B^p.Ser526Pro^ depicted that Gly592, Ile834, and Thr544 exhibited 9.91 Å, 9.7, and 9.2 Å. The flexibility analysis depicted variable residues within the Loop and C-terminal regions contributing more to the overall flexibility with variable intensities (Figure 5).

#### 3.8.3. Gyration Analysis

The radius of gyration analysis was used to analyze parameters such as overall dimensions and the compaction level of the molecules (Figure 6). The Rg value of the natural protein was about 50 at the start of the simulation. During the experiment, this number fluctuated between 43.5 Å and 52.4 Å, eventually reaching 48.2 Å at the conclusion. The GluN2B^p.Thr685Pro^ and S526P mutations have considerably lower Rg values than the natural protein. The Rg values of the GluN2B^p.Asn615Ile^ and GluN2B^p.Arg682Cys^ variants were greater than the native protein values throughout the simulation period, except for 80–85 ns when the Rg value of GluN2B^p.Asn615Ile^ fluctuated lower than that of GluN2B^WT^. Furthermore, the Rg values of GluN2B^p.Asn615Ile^ and GluN2B^p.Arg682Cys^ exhibited similarly convergent Rg scores throughout the trajectory. According to the Rg study, proteins with the variations showed a greater amount of variable compaction than the wild-type protein.

#### 3.8.4. Intramolecular Hydrogen Bond Comparison of Wildtype and Mutant Variants

Hydrogen bond analysis is critical for understanding the intramolecular hydrogen bond network of GluN2B protein, both natural and mutant. Figure 7 depicts the hydrogen bond network of all systems in detail. A careful examination of the number of hydrogen bonds found that the number of hydrogen bonds greatly changed amongst all of the systems after point mutation. The number of hydrogen bonds in GluN2B^WT^ was 728 at the start of the simulation and decreased to 710 at the conclusion. In contrast, the GluN2B^p.Asn615Ile^ protein had 722 at the start and grew to 734 at the end of the simulation period. At the start of the simulation, GluN2B^p.Thr685Pro^ had 729 hydrogen bonds, which fell to 692 at the conclusion. GluN2B^p.Arg682Cys^ showed 749 at the start and 708 at the conclusion of the simulation, while GluN2B^p.Ser526Pro^ showed 732 at the start and 739 at the end. The average number of hydrogen bonds varied across all systems. GluN2B^WT^ and GluN2B^p.Asn615Ile^, GluN2B^p.Thr685Pro^, GluN2B^p.Arg682Cys^, and GluN2B^p.Ser526Pro^ have 718.2, 729.9, 715.0, 711.0, and 726.2 hydrogen bonds on average. This research reveals that the hydrogen bond occupancy has shifted dramatically.

#### 3.8.5. Secondary Structure Elements Comparison over the Trajectory

Secondary structure elements (SSE) that contribute to the overall protein stability were also analyzed for both systems, and it was discovered that the GluN2B^WT^ complex was maintaining an average of approximately 48.82 percent SSE. The majority of the secondary structure elements were composed of helices (33.66%), rather than strands (15.17%). On the other hand, the p.Asn615Ile displayed an overall percentage of SSE that was 49.93 percent, with helices making up 34.56 percent and strands making up 15.36 percent. p.Thr685Pro exhibited a total SSE of 49.19% with helix 34.24% and strand 14.95%, while p.Arg682Cys exhibited 48.32% SSE with helices counting for 33.17% and strands contributing 15.16%. p.Ser526Pro has 50.05% of SSE with helices counting for 35.29% and strands accounting for 14.76%. When we further investigated why the GluN2B variant posed less stable and variable conformations, we discovered that the GluN2B residues at the end of the SSE graph (Figure 8) were transitioning from strands to loops. This resulted in the loss of GluN2B SSE elements, which likely harmed its overall stability and conformational condition.

### 3.9. Comparison of GluN2B^WT^ and GluN2B^MT^

The four polymorphisms that were previously indicated as harmful by all of the prediction algorithms demonstrated that the structure of the native protein changes when compared to the native protein. Furthermore, Figure 9 depicts GluN2B^WT^ and four variations, GluN2B^p.Asn615Ile^, GluN2B^p.Thr685Pro^, GluN2B^p.Arg682Cys^, and GluN2B^p.Ser526Pro^, overlaid at various timelines. Mutations alter the creation of coiling and the angles between folding, resulting in a change in the overall structure of proteins. To determine the modifications, the mutant structures were overlaid on the original structures using a structural alignment tool, which demonstrated the change in 3D domain development at the location of the mutation (Figure 9). The superimposition caused major structural changes in the loop and C-terminal regions. The different overlapping structures may be observed clearly, while the mutational alteration contributes to the overall structural behavior of the protein. The change in structure also causes a change in function, which may influence the protein’s active sites, ion channels, and binding sites.

### 3.10. Functional Displacement of GluN2B and Mutant Models

We used the dynamics cross-correlation matrix (DCCM) approach to determine the functional displacement of all systems as a function of time. The findings demonstrate that the GluN2B residues at the N and C terminals correlated positively. Mutant GluN2B^p.Arg682Cys^ exhibited a similar correlation to the GluN2B^WT^, confirming that the positive correlation may be attributable to the adopted confirmation of these two proteins. Furthermore, additional mutants such as GluN2B^p.Asn615Ile^, GluN2B^p.Thr685Pro^, and GluN2B^p.Ser526Pro^ demonstrated a stronger positive association than GluN2B^WT^ (Figure 10). Overall, the DCCM data reveal that the natural protein and mutant protein exhibit distinct patterns of significant positive association. The deep blue tint shows a high negative association among the residues, whereas the red color suggests a strong positive correlation. Positively connected residues migrate in the same direction, whereas negatively correlated residues move oppositely.

### 3.11. Dimensionality Reduction Using PCA

Motion mode analysis was used to investigate dynamically favorable conformational changes in the chemistry of GluN2B^WT^ and four variations, GluN2B^p.Asn615Ile^, GluN2B^p.Thr685Pro^, GluN2B^p.Arg682Cys^, and GluN2B^p.Ser526Pro^. The coordinate covariance matrix computed from the time series of 3D positional coordinates of distinct variant models during the 100 ns MD simulation duration was chosen as the input for principal component analysis (PCA). The findings demonstrate that all of the systems, GluN2B^WT^ and four variations, GluN2B^p.Asn615Ile^, GluN2B^p.Thr685Pro^, GluN2B^p.Arg682Cys^, and GluN2B^p.Ser526Pro^, had diverse patterns and did not converge from one energy state to another, indicating that the mutant states of the GluN2B protein had an unstable pattern of conformations (Figure 11).

## 4. Discussion

Conventional strategies for finding nucleotide changes in genes and their influence on the associated protein in vitro are not only difficult but also slow and arduous to apply. To make things easier, investigations based on computational biology may be carried out [57,58]. SNP stands for single nucleotide polymorphism, and it is the foundation of genetic diversity [24,25,59]. The vast majority of SNPs are unimportant. However, certain so-called nonsynonymous SNPs (nsSNPs) may affect a gene, predisposing humans to a variety of disorders. Since its discovery, genetic variations in the *GRIN2B* translating into GluN2B protein have consistently been linked to West syndrome, intellectual impairment with focal epilepsy, developmental delay, macrocephaly, corticogenesis, brain plasticity, as well as infantile spasms and Lennox–Gastaut syndrome [5,60,61,62,63]. It is unknown, however, how *GRIN2B* genetic variation impacts protein function.

The goal of this computational research is to use prediction methods to examine the effects of nonsynonymous SNPs on protein structure and function. The dbSNP database was used to gather nsSNPs. These were put through thirteen prediction algorithms. At least 9 different computer programs found that 19 mutations in the *GRIN2B* encoding protein (p.Gly820Val, p.Gly820Ala, p.Gly820Glu, p.Gly820Arg, p.Ile751Thr, p.Arg696His, p.Ile695Ser, p.Ile695Thr, p.Thr685Pro, p.Arg682Cys, p.Asp668Tyr, p.Asp668Asn, p.Ala652Pro, p.Ala639Val, p.Val618Gly, p.Asn615Ile, p.Trp607Cys, p.Ser526Pro, and p.Alu413Gly) were the most harmful nsSNPs. The disparity in the findings of the prediction software employed in this work highlighted the need to employ more than one method to evaluate the influence of alterations on the structure and function of the protein [64].

The conservation analysis findings revealed that all these nsSNPs were found in protected areas. Indeed, the highly conserved amino acids, according to Miller and Kumar, are found in physiologically active places. The biological activities are changed when these residues are replaced [65]. After YASARA software visualization, the eleven SNPs identified as pathogenic revealed a loss of hydrogen and hydrophobic bonds compared to the wild-type protein. Indeed, Wang and Moult demonstrated that nearly 80% of their nsSNP-related diseases generated a destabilization protein [66].

It should be highlighted that the p.Ser526Pro variant was found in the S1 domain, p.Asn615Ile in pore while p.Arg682Cys and p.Thr685Pro were found in the S2 domain; S1 and S2 form the ligand-binding domain, pore; re-entrant pore-forming and transmembrane spanning domains, which consequently may account for the change in spatial conformation and, as a result, instability of the “GluN2B subunit of N-methyl-D-aspartate receptors” complex, which may be a beneficial risk factor for neurodevelopmental disorders. As a result of harmful SNPs, loss or gain of hydrogen bonds, hydrophobic interactions, and salt bridges may change how proteins are built and how they work [24,67,68].

Bioinformatics techniques now use simulation to investigate the various effects of protein mutations [24]. Molecular dynamics models reproduce the genuine behavior of molecules in their surroundings [69]. This computational platform gives more specific information on particle motion, stability, flexibility, and overall protein dimensions as they change over time [70]. Furthermore, this powerful analysis has the strongest association with experimental investigations [71,72]. In reality, these characteristics are interconnected; when investigating protein structure, they must be investigated concurrently, and variations that impair one property of a protein may have a direct impact on the other [73,74,75].

RMSD analysis showed that the changes to the proteins GluN2B^p.Asn615Ile^, GluN2B^p.Thr685Pro^, GluN2B^p.Arg682Cys^, and GluN2B^p.Ser526Pro^ contributed more variable instability to the GluN2B native protein.

These harmful SNPs may cause the most damage to protein stability. According to Yue and Moult, 25% of pathogenic SNPs in the human population may affect protein function via protein stability changes [76,77]. Furthermore, several investigations have demonstrated that decreased protein stability causes an increase in protein breakdown, aggregation, and misfolding [25,78]. The stability of the protein was altered by the mutations GluN2B^p.Asn615Ile^, GluN2B^p.Thr685Pro^, GluN2B^p.Arg682Cys^, and GluN2B^p.Ser526Pro^. Studies of evolutionary stability and mutations that affect genes that code for proteins have shown that leucine, serine, and arginine are the amino acids that most affect the stability of proteins in mutants [25,79].

The RMSF calculations in our research revealed that the 11 nsSNPs affect flexibility at distinct levels. Flexibility is an important feature of protein function. It enables proteins to respond to environmental and chemical changes. Molecular flexibility regulates various processes, including enzymatic catalysis and protein activity regulation. Indeed, a change that impacts protein flexibility has the potential to interfere with their function and, as a result, disease development [80,81,82]. Rg represents the molecule’s overall spread and is calculated as the root mean square distance of atoms collected from their common center of gravity. Indeed, Lobanov et al. show that the radius of gyration is a good predictor of protein structure compactness [83]. In this study, the Rg analysis showed that the variants GluN2B^p.Asn615Ile^, GluN2B^p.Thr685Pro^, and GluN2B^p.Ser526Pro^ were more tightly packed than the wild-type protein except for GluN2B^p.Arg682Cys^. These results were further validated by DCCM, PCA, and conformational analysis.

According to the following research, we discovered four pathogenic variants among our findings that were shown to be involved in disease pathogenesis. Lemke et al. conducted studies on two people with West syndrome and significant developmental delay, as well as one person with ID and focal epilepsy. The ID patient had a missense mutation in the extracellular glutamate-binding domain (p.Arg540His), while both West syndrome patients had missense mutations in the NR2B ion channel-forming re-entrant loop (p.Asn615Ile, p.Val618Gly). Subsequent testing of 47 individuals with unexplained infantile spasms revealed no new de novo mutations [3]. p.Thr685Pro is also linked with developmental and epileptic encephalopathy, 27 as submitted to the NCBI by HudsonAlpha Institute for Biotechnology, CSER-HudsonAlpha (Accession no: SCV000265519). p.Arg682Cys was reported by multiple groups to be associated with autosomal dominant 6, intellectual developmental disorder [84]. p.Ser526Pro was associated with the complex neurodevelopmental disorder as explained in the submission from Simons Searchlight facilitated by GenomeConnect (Accession: SCV001443616.1). These results also reiterate the methodology followed by our research to be impactful in identifying the most deleterious SNPs.

## 5. Conclusions

This is the first comprehensive in silico examination of functional SNPs in the *GRIN2B* gene. Because of their presence in a highly conserved area and capacity to impact protein stability, we identified 19 nsSNPs (p.Gly820Val, p.Gly820Ala, p.Gly820Glu, p.Gly820Arg, p.Ile751Thr, p.Arg696His, p.Ile695Ser, p.Ile695Thr, p.Thr685Pro, p.Arg682Cys, p.Asp668Tyr, p.Asp668Asn, p.Ala652Pro, p.Ala639Val, p.Val618Gly, p.Asn615Ile, p.Trp607Cys, p.Ser526Pro, p.Alu413Gly) as potentially harmful that may change the structure and/or function of the *GRIN2B* encoded N-methyl D-aspartate receptor subtype 2B protein. The detailed analysis of our study’s findings highlighted the significance of variation at the Thr685, Arg682, Asn615, and Ser526 positions involved in post-translational modifications directly or indirectly, which may be at the root of the destabilization of the GluN2B and, as a result, the occurrence of the disease. p.Thr685Pro, p.Arg682Cys, p.Asn615Ile, and p.Ser526Pro were identified as pathogenic, with the potential to inflict significant functional and stability impacts on the proteins. The outcomes of this research are succinct evidence that these findings might serve as a baseline for possible diagnostic and therapeutic approaches.

## Figures and Tables

**Figure 1 genes-13-01332-f001:**
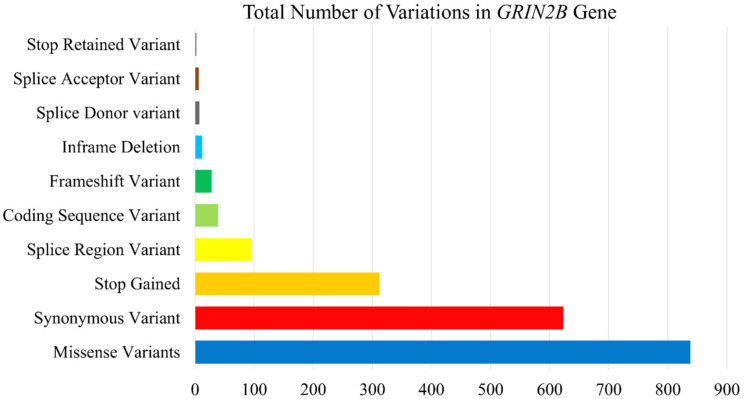
Classification of the total number of variations in the *GRIN2B* based on their locus.

**Figure 2 genes-13-01332-f002:**
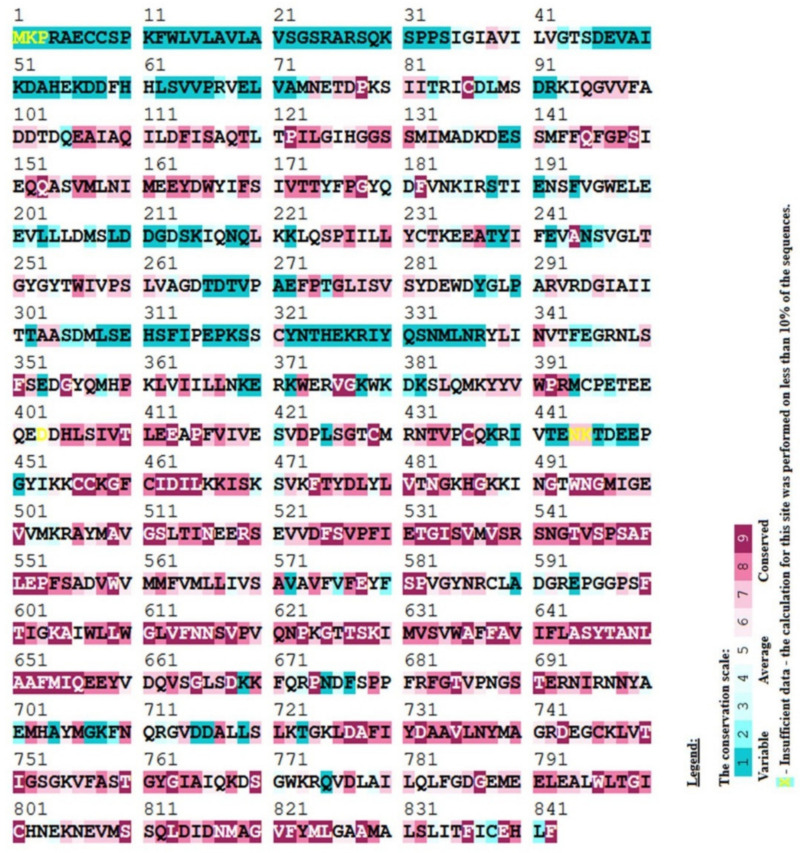
Conservation analysis by ConSurf server.

**Figure 3 genes-13-01332-f003:**
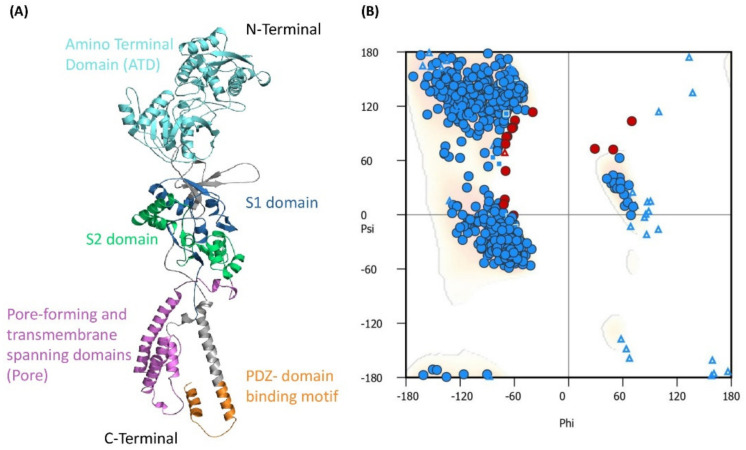
(**A**) The three-dimensional structure of GluN2B protein is classified by color-coded domains and motifs. (**B**) Validation of the 3D structure using Ramachandran Plot. The red color indicates the residues in unfavored region while blue color indicates residues in favored region.

**Figure 4 genes-13-01332-f004:**
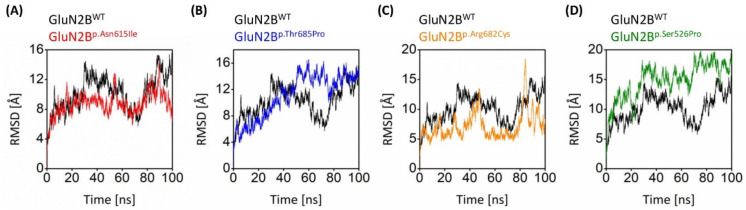
RMSD plot manifesting the collective conformational drifts in both GluN2B^WT^ and GluN2B^MT^ models. RMSD comparison of native GluN2B with (**A**) GluN2B^p.Asn615Ile^, (**B**) GluN2B^p.Thr685Pro^, (**C**) GluN2B^p.Arg682Cys^, and (**D**) GluN2B^p.Ser526Pro^.

**Figure 5 genes-13-01332-f005:**
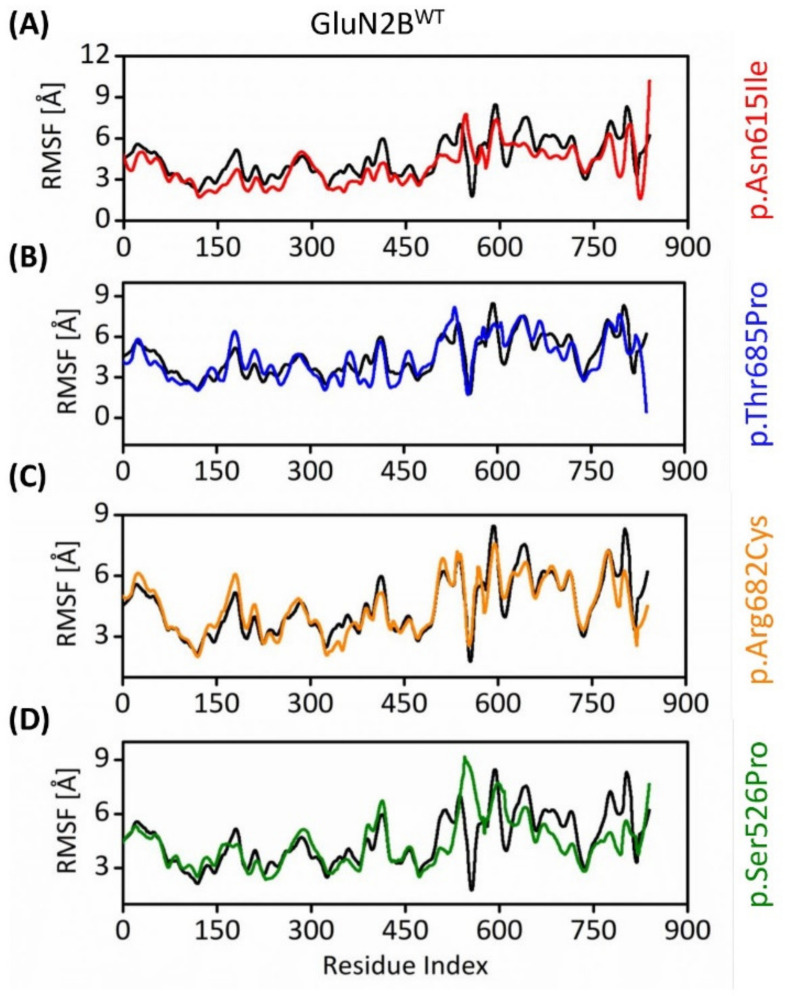
RMSF plot manifesting the collective flexibility of GluN2B^WT^ protein model with (**A**) GluN2B^p.Asn615Ile^, (**B**) GluN2B^p.Thr685Pro^, (**C**) GluN2B^p.Arg682Cys^, and (**D**) GluN2B^p.Ser526Pro^. The black color in the line graph represents the RMSF of GluN2B^WT^.

**Figure 6 genes-13-01332-f006:**
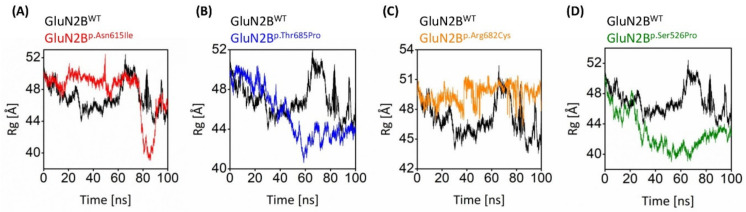
Black-colored lines in all graphs show the plot of the native GluN2B. Rg graphs of (**A**) GluN2B^p.Asn615Ile^, (**B**) GluN2B^p.Thr685Pro^, (**C**) GluN2B^p.Arg682Cys^, and (**D**) GluN2B^p.Ser526Pro^.

**Figure 7 genes-13-01332-f007:**
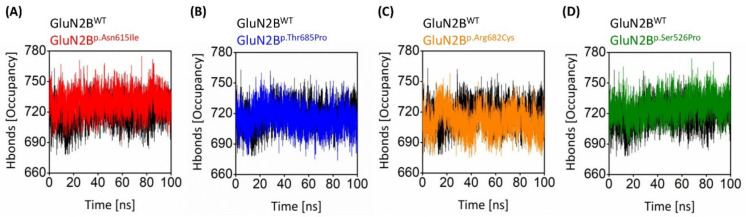
The number of hydrogen bonds in the three-dimensional structure displayed versus the amount of time spent running the simulation. The GluN2B^WT^ is shown in comparison with (**A**) GluN2B^p.Asn615Ile^, (**B**) GluN2B^p.Thr685Pro^, (**C**) GluN2B^p.Arg682Cys^, and (**D**) GluN2B^p.Ser526Pro^.

**Figure 8 genes-13-01332-f008:**
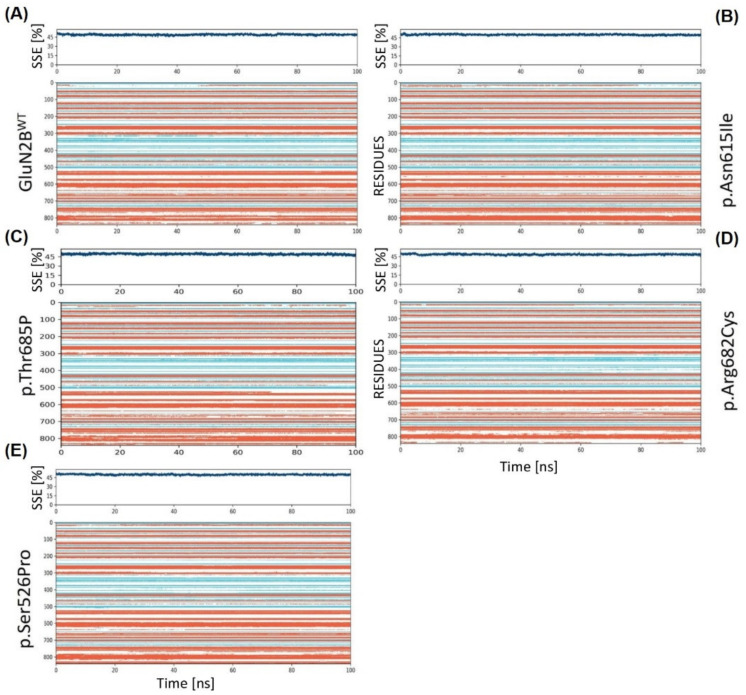
The graphic highlights the SSE composition for each trajectory frame during the simulation and displays SSE distribution by residue index across the protein structure. Over a 100 ns simulation duration, the graphic tracks each residue and its SSE assignment. (**A**) GluN2B^WT^ residue index and SSE with the y-axis, (**B**) GluN2B^p.Asn615Ile^ SSE, (**C**) GluN2B^p.Thr685Pro^ SSE, (**D**) GluN2B^p.Arg682Cys^ SSE, and (**E**) GluN2B^p.Ser526Pro^ SSE. The red color represents α-helices while cyan color depicts β-strands.

**Figure 9 genes-13-01332-f009:**
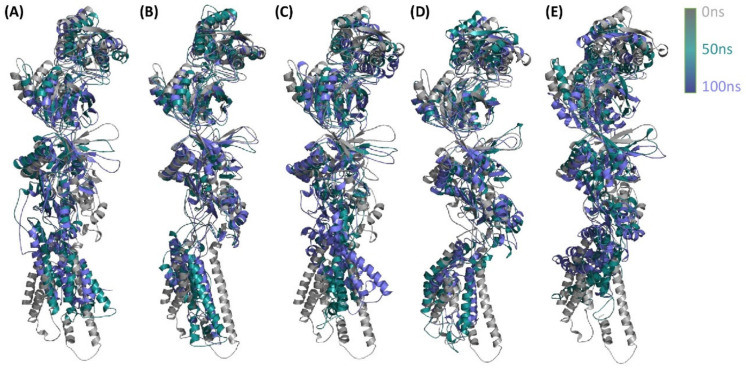
The RMSDs of GluN2B^WT^ and GluN2B^MTs^ at different simulation periods. The GluN2B^WT^ behaves very differently from the GluN2B^MTs^. The mutant models (GluN2B^p.Asn615Ile^, GluN2B^p.Thr685Pro^, GluN2B^p.Arg682Cys^, and GluN2B^p.Ser526Pro^) are shown in (**A**), (**B**), (**C**), (**D**) and (**E**) respectively.

**Figure 10 genes-13-01332-f010:**
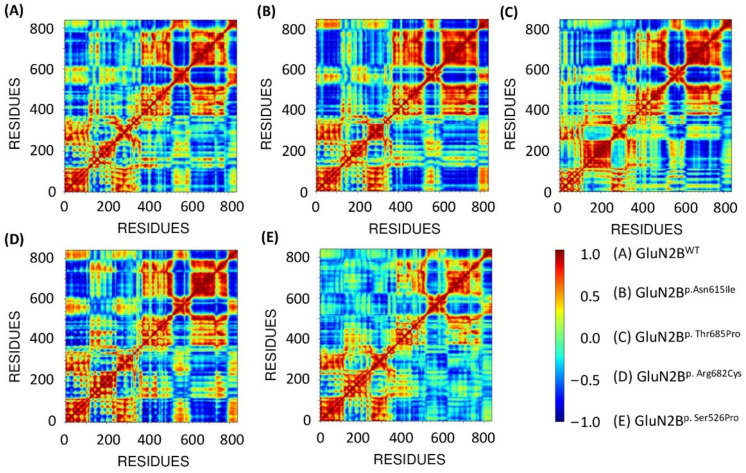
DCCM of the native protein GluN2B^WT^ (**A**) and four variants GluN2B^p.Asn615Ile^ (**B**), GluN2B^p.Thr685Pro^ (**C**), GluN2B^p.Arg682Cys^ (**D**), and GluN2B^p.Ser526Pro^ (**E**).

**Figure 11 genes-13-01332-f011:**
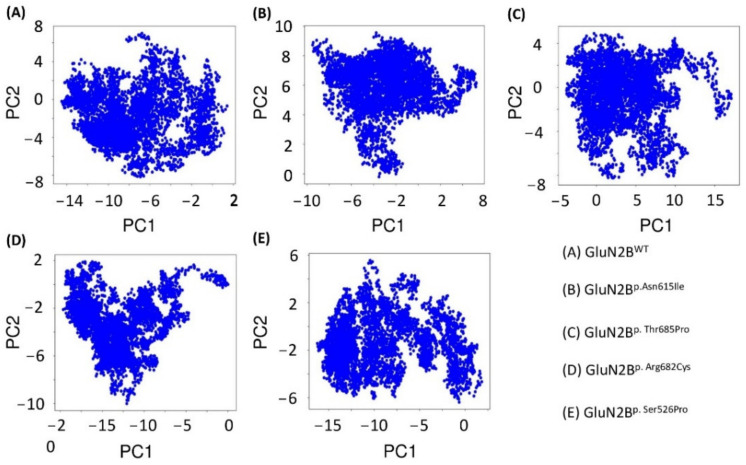
PCA of the native protein GluN2B^WT^ (**A**) and four variants GluN2B^p.Asn615Ile^ (**B**), GluN2B^p.Thr685Pro^ (**C**), GluN2B^p.Arg682Cys^ (**D**), and GluN2B^p.Ser526Pro^ (**E**).

**Table 1 genes-13-01332-t001:** Impact of changes in *GRIN2B* due to amino acid substitution predicted by SIFT, Polyphen2, SNAP2, CADD, REVEL, MetaLR, and Mutation Assessor.

Variant ID	Mutation	SIFT	Polyphen2	SNAP2	CADD	REVEL	MetaLR	Mut. Ass.
rs876661167	p.Arg1111His *	deleterious	benign	neutral	likely benign	likely benign	tolerated	medium
rs1064794979	p.Gly826Glu *	tolerated	possibly damaging	effect	likely benign	likely benign	tolerated	medium
rs797044849	p.Gly820Val	deleterious	probably damaging	effect	likely benign	likely disease-causing	tolerated	medium
rs797044849	p.Gly820Ala	deleterious	probably damaging	effect	likely benign	likely disease-causing	tolerated	high
rs797044849	p.Gly820Glu	deleterious	probably damaging	effect	likely benign	likely disease-causing	tolerated	high
rs1555103150	p.Gly820Arg	deleterious	probably damaging	effect	likely benign	likely disease-causing	tolerated	high
rs864309560	p.Ser810Arg *	deleterious	probably damaging	effect	likely benign	likely benign	tolerated	medium
rs876661055	p.Ile751Thr	deleterious	probably damaging	effect	likely benign	likely disease-causing	tolerated	medium
rs1555103971	p.Arg696His	deleterious	probably damaging	effect	likely benign	likely disease-causing	tolerated	medium
rs876661219	p.Ile695Ser	deleterious	probably damaging	effect	likely deleterious	likely disease-causing	tolerated	medium
rs876661219	p.Ile695Thr	deleterious	probably damaging	effect	likely benign	likely disease-causing	tolerated	medium
rs869312868	p.Gly689Ser *	deleterious	probably damaging	effect	likely benign	likely benign	tolerated	low
rs869312669	p.Thr685Pro	deleterious	probably damaging	effect	likely benign	likely disease-causing	tolerated	high
rs387906636	p.Arg682Cys	deleterious	probably damaging	effect	likely deleterious	likely disease-causing	tolerated	medium
rs876661151	p.Asp668Tyr	deleterious	probably damaging	effect	likely benign	likely disease-causing	damaging	high
rs876661151	p.Asp668Asn	deleterious	probably damaging	effect	likely benign	likely disease-causing	damaging	high
rs1555110812	p.Ala652Pro	deleterious	probably damaging	effect	likely benign	likely disease-causing	damaging	high
rs797044930	p.Ala639Val	deleterious	probably damaging	effect	likely benign	likely disease-causing	tolerated	medium
rs672601376	p.Val618Gly	deleterious	probably damaging	effect	likely benign	likely disease-causing	tolerated	medium
rs672601377	p.Asn615Ile	deleterious	probably damaging	effect	likely benign	likely disease-causing	tolerated	medium
rs1057518700	p.Trp607Cys	deleterious	probably damaging	effect	likely benign	likely disease-causing	damaging	high
rs1057519004	p.Val558Ile *	deleterious	possibly damaging	effect	likely benign	likely benign	tolerated	low
rs1131691702	p.Ser526Pro	deleterious	possibly damaging	effect	likely benign	likely disease-causing	tolerated	medium
rs886041295	p.Asn516Ser *	deleterious	benign	effect	likely benign	likely benign	tolerated	low
rs527236034	p.Glu413Gly	deleterious	probably damaging	effect	likely deleterious	likely disease-causing	tolerated	medium

Remarks: Asterisk (*): Mutations that were annotated as benign or likely tolerated by more than two tools. These mutations are excluded from further study.

**Table 2 genes-13-01332-t002:** Disease association of the SNPs as predicted by PhD-SNP, SNPs & GO, I-Mutant, and MuPro.

Variant ID	Mutation	PhD-SNP	SNPs & GO	I-Mutant	MUpro
rs797044849	p.Gly820Val	Disease	Disease	−0.34 (Dec. Stab.)	−0.86017491 (Dec. Stab.)
rs797044849	p.Gly820Ala	Disease	Disease	−0.55 (Inc. Stab.)	−1.2722781 (Dec. Stab.)
rs797044849	p.Gly820Glu	Disease	Disease	−0.55 (Dec. Stab.)	−0.96468908 (Dec. Stab.)
rs1555103150	p.Gly820Arg	Disease	Disease	−0.41 (Dec. Stab.)	−1.0760922 (Dec. Stab.)
rs876661055	p.Ile751Thr	Neutral	Disease	−2.27 (Dec. Stab.)	−2.8133707 (Dec. Stab.)
rs1555103971	p.Arg696His	Disease	Disease	−1.27 (Dec. Stab.)	−1.8584507 (Dec. Stab.)
rs876661219	p.Ile695Ser	Disease	Disease	−2.16 (Dec. Stab.)	−1.876043 (Dec. Stab.)
rs876661219	p.Ile695Thr	Neutral	Disease	−2.11 (Dec. Stab.)	−2.047267 (Dec. Stab.)
rs869312669	p.Thr685Pro	Disease	Disease	−0.92 (Dec. Stab.)	−1.1060293 (Dec. Stab.)
rs387906636	p.Arg682Cys	Disease	Disease	−1.11 (Dec. Stab.)	−0.790331 (Dec. Stab.)
rs876661151	p.Asp668Tyr	Disease	Disease	−0.58 (Dec. Stab.)	−0.84414843 (Dec. Stab.)
rs876661151	p.Asp668Asn	Neutral	Disease	−1.42 (Dec. Stab.)	−1.3280612 (Dec. Stab.)
rs1555110812	p.Ala652Pro	Disease	Disease	−0.30 (Dec. Stab.)	−1.8518457 (Dec. Stab.)
rs797044930	p.Ala639Val	Disease	Disease	−0.16 (Dec. Stab.)	−0.89362766 (Dec. Stab.)
rs672601376	p.Val618Gly	Disease	Disease	−2.62 (Dec. Stab.)	−2.1850285 (Dec. Stab.)
rs672601377	p.Asn615Ile	Disease	Disease	0.78 (Inc. Stab.)	−0.5777985 (Dec. Stab.)
rs1057518700	p.Trp607Cys	Disease	Disease	−1.73 (Dec. Stab.)	−1.1378457 (Dec. Stab.)
rs1131691702	p.Ser526Pro	Disease	Disease	−0.32 (Inc. Stab.)	−1.5731478 (Dec. Stab.)
rs527236034	p.Glu413Gly	Neutral	Disease	−1.25 (Dec. Stab.)	−1.1247393 (Dec. Stab.)

**Table 3 genes-13-01332-t003:** Conservation Analysis, PTMs, predicted PTM sites, and peptides.

Variant ID	Mutation	ConSurf	Musite	Findmod Peptides	Potential Modification
rs797044849	p.Gly820Val	9, bur, str	---	---	---
rs797044849	p.Gly820Ala	9, bur, str	---	---	---
rs797044849	p.Gly820Glu	9, bur, str	---	---	---
rs1555103150	p.Gly820Arg	9, bur, str	---	---	---
rs876661055	p.Ile751Thr	7, exp	---	---	---
rs1555103971	p.Arg696His	7, exp	---	---	---
rs876661219	p.Ile695Ser	9, bur, str	---	---	---
rs876661219	p.Ile695Thr	9, bur, str	---	---	---
rs869312669	p.Thr685Pro	8, exp, fun	Phosphorylation	---	---
rs387906636	p.Arg682Cys	5, exp	Methylation	FQRPNDFSPPFR	DHAS
rs876661151	p.Asp668Tyr	9, exp, fun	---	---	---
rs876661151	p.Asp668Asn	9, exp, fun	---	---	---
rs1555110812	p.Ala652Pro	9, bur, str	---	---	---
rs797044930	p.Ala639Val	9, bur, str	---	---	---
rs672601376	p.Val618Gly	8, bur	---	---	---
rs672601377	p.Asn615Ile	8, exp, fun	Glycosylation	---	---
rs1057518700	p.Trp607Cys	8, bur	---	---	---
rs1131691702	p.Ser526Pro	9, bur, str	---	SEVVDFSVPFIETGISVMVSR	BROM
rs527236034	p.Alu413Gly	8, exp, fun	---	---	---

Abbreviations: DHAS; 2,3-didehydroalanine (Ser), BROM; Bromination: Light blue color: Potential Residues involved.

**Table 4 genes-13-01332-t004:** Summary of the most deleterious SNPs.

Variant ID	Mutation	Functional Impact	Conservation	PTMs (Residual)	PTM Driver Peptides	Pep. Potential PTMs
rs869312669	p.Thr685Pro	Disease	8, exp, fun	Phosphorylation	---	---
rs387906636	p.Arg682Cys	Disease	5, exp	Methylation	FQRPNDFSPPFR	2,3-didehydroalanine (Ser)
rs672601377	p.Asn615Ile	Disease	8, exp, fun	Glycosylation	---	---
rs1131691702	p.Ser526Pro	Disease	9, bur, str	---	SEVVD FSVPFIETGISVMVSR	Bromination

Light blue color: Potential Residues involved.

## Data Availability

All data generated or analyzed during the study are included in this article.

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
