# Peer review of "Molecular Insights into the Role of Pathogenic nsSNPs in GRIN2B Gene Provoking Neurodevelopmental Disorders"

_genes, 2022, doi:10.3390/genes13081332_

Round 1

Reviewer 1 Report

The Authors aim to disclose how the pathogenetic variations in the GRIN2B gene cause the associated clinical features. They face their aim using an in silico approach. Even if this might be an innovative study, there are major criticisms.

The first part of the manuscript is very confusing and redundant. The Authors make great confusion in the variant classification. What do they mean for coding variants? Because missense, frameshift, in-frame, stop gained and synonymous are coding variants! The nomenclature of the variants is also wrong; they should adopt the HGVS nomenclature.

The need for several prediction tools to achieve a good interpretation of a nucleotide variant in a disease gene is an acquired concept, so the advice is to remove the first part from the manuscript.

The simulation for structure prediction and the solvent molecular Dynamics of pathogenic variants are well done and may give some cues for functional studies.

Reviewer 2 Report

This is a very valuable comprehensive computational study complimenting the genetic findings. The authors aimed to determine the cumulative pathogenic impact of known single nucleotide polymorphisms in the GRIN2B gene, as well as their associations with neurodevelopmental disorders and the influence of deleterious nsSNPs on protein structural behaviour using different bioinformatic tools.

This analysis identified 19 nsSNPs as potentially harmful that may change the structure and/or function of the GRIN2B encoded N-methyl D-aspartate receptor subtype 2B protein. Among those four polymorphisms (rs869312669 (T685P), rs387906636 (R682C), rs672601377 (N615I), and rs1131691702 (S526P)) were identified as pathogenic causing destabilization of the GluN2B.

The study covers some issues that reveal how GRIN2B genetic variation impacts protein function. The authors suggest the relevance of GRIN2B genetic variations in clinical significance of neurodevelopmental disorders. The structure of the manuscript appears adequate and well divided into the sections. Moreover, the study is easy to follow. The analysis is well-designed and carefully performed. I have no negative comments.

Reviewer 3 Report

The manuscript titled, "Molecular insights into the role of pathogenic nsSNPs in GRIN2B Gene provoking neurodevelopmental disorders by Shah et al., is an interesting manuscript. Using computational approaches, the authors have tried to identify the pathogenic variations in GRIN2B gene. 

One of the major limitation of the present study is the lack of proper controls.

1) It would be very important to compare the pathogenic variant predictions for GRIN2B with a known gene with known pathogenic variants that have been identified through computational approaches and atleast few of them have been validated in molecular and diagnostic setting. Maybe cite those references where this approach was used and further studies were done.

2)Similarly, a negative control is also needed in these studies. 

3) Better description of parameters that have been chosen for modeling these variants is needed. The manuscript currently lacks robust description of parameters. 
